# Three-dimensional Pentagon Carbon with a genesis of emergent fermions

Chengyong Zhong[1], Yuanping Chen[1], Zhi-Ming Yu[2], Yuee Xie[1], Han Wang[3], Shengyuan A. Yang[2] & Shengbai Zhang[3]

Carbon, the basic building block of our universe, enjoys a vast number of allotropic structures. Owing to its bonding characteristic, most carbon allotropes possess the motif of hexagonal rings. Here, with first-principles calculations, we discover a new metastable three-dimensional carbon allotrope entirely composed of pentagon rings. The unique structure of this Pentagon Carbon leads to extraordinary electronic properties, making it a cornucopia of emergent topological fermions. Under lattice strain, Pentagon Carbon exhibits topological phase transitions, generating a series of novel quasiparticles, from isospin-1 triplet fermions to triply degenerate fermions and further to Hopf-link Weyl-loop fermions. Its Landau level spectrum also exhibits distinct features, including a huge number of almost degenerate chiral Landau bands, implying pronounced magneto-transport signals. Our work not only discovers a remarkable carbon allotrope with highly rare structural motifs, it also reveals a fascinating hierarchical particle genesis with novel topological fermions beyond the Dirac and Weyl paradigm.

[1] School of Physics and Optoelectronics, Xiangtan University, Xiangtan, Hunan 411105, China. [2] Research Laboratory for Quantum Materials, Singapore University of Technology and Design, Singapore 487372, Singapore. [3] Department of Physics, Applied Physics, and Astronomy, Rensselaer Polytechnic Institute, Troy, New York 12180, USA. Correspondence and requests for materials should be addressed to Y.C. (email: chenyp@xtu.edu.cn) or to S.A.Y. (email: shengyuan_yang@sutd.edu.sg).

M ost stable carbon allotropes, including the prominent members such as graphite, diamond, carbon nanotube[1] and graphene[2], possess a motif of hexagonal rings due to the dominating $sp^2$ and $sp^3$ bonding character[3]. In contrast, the motif of pentagonal rings, although can appear in molecules such as fullerene[4], is very rare in three-dimensional (3D) allotropes. This is partly due to the unfavourable bonding angles that may easily cause structural instability[5], and also because the pentagonal motif does not readily fit into the 3D space group symmetry[6]. It seems unlikely to have a stable 3D carbon allotrope dominated by pentagonal carbon rings.

Meanwhile, the research on topological metals and semimetals (TMs) has been attracting tremendous interest[7–14]. In these materials, topologically protected quasiparticles emerge at protected band-crossing points around Fermi Level, giving rise to condensed matter realizations of many peculiar fermions with a variety of distinct electronic properties. For example, in Dirac and Weyl semimetals, the quasiparticles near the protected linear band-crossing points are direct analogues of the Dirac and Weyl fermions in relativistic quantum field theory, providing unique opportunities to study interesting high-energy physics phenomena such as chiral anomaly and gravitational anomaly in a table-top experiment[15–18]. The emergent topological fermions can be classified by the degeneracy of the band-crossing point, which dictates the number of internal degrees of freedom of the fermion. Dirac and Weyl fermions are associated with crossing points with fourfold and twofold degeneracies[19,20]. Importantly, the breaking of Lorentz symmetry at lattice scale makes it possible to have TMs that can host new types of fermions with different degeneracies beyond the Dirac and Weyl fermions[21–23]. These new fermions are now actively searched for, and if realized, they are believed to host a wealth of exotic physical properties unprecedented in high-energy physics.

The search for TMs has been focused on compounds involving heavy elements[7,24,25], hoping that the strong spin–orbit coupling could help to achieve a nontrivial band structure. On the other hand, light elements such as carbon may offer a distinct alternative. For example, graphene is well-known with two-dimensional (2D) Dirac fermions[26]. With negligible spin–orbit coupling, the real spin in carbon materials can be considered as a dumb degree of freedom[27]. Then the fundamental time reversal operation ($\mathcal{T}$) here satisfies $\mathcal{T}^2 = 1$, contrasting with $\mathcal{T}^2 = -1$ for the spinful case. Thus in terms of topological classifications, the nontrivial phases in carbon would be fundamentally distinct from those in spin–orbit-coupled systems. Indeed, previous studies have identified a few TM carbon allotropes with Weyl points[28,29], nodal loops[27,29–32] and even Weyl surfaces[28]. Motivated by these observations, one may naturally wonder: is it possible to also realize new types of topological fermions in carbon allotropes?

In this work, based on first-principles calculations, we propose a new 3D carbon allotrope, named as 3D Pentagon Carbon (3D-C5), which is entirely composed of pentagonal carbon rings. With nicely interlaced structures, the structure shows good stability comparable to other metastable carbon allotropes. We characterize its phononic, mechanical and electronic properties. At the equilibrium state, Pentagon Carbon is a narrow-gap semiconductor. A semiconductor–metal quantum phase transition can be driven by an applied strain. Remarkably, at the critical transition point, three bands cross at a single Fermi point in the energy-momentum space, making the emergent low-energy fermions a novel type of isospin-1 triplet fermions. Beyond the transition, the single Fermi point splits into two triply degenerate band-crossing points along the fourfold screw-rotational axis. The material is turned into a novel TM possessing

another type of triply degenerate fermions. When further breaking the screw-rotational symmetry, the triply degenerate fermion points will transform into two inter-connected Hopf-link Weyl loops protected by the remaining mirror symmetry. We show that, for each case, the Landau level (LL) spectrum exhibits distinct features tied to the topological fermions, suggesting pronounced anomalies in magneto-transport experiment. In addition, we discuss a possible reaction pathway for the experimental synthesis of Pentagon Carbon.

## Results

**Lattice structure and stability.** The structure of 3D Pentagon Carbon is shown in Fig. 1a , which can be viewed as formed by interlinking two orthogonal arrays of pentagon-ring nanoribbons and stacking them along the z-direction (Fig. 1b). After linking, the two nanoribbons share one (green-colored) atom. Hence in the Pentagon Carbon lattice, there are two kinds of carbon atoms, that is, the $sp^2$-hybridized (red- and blue-colored) atoms forming the backbones of the pentagon-ring nanoribbons and the $sp^3$-hybridized (green-colored) atoms forming the linkage between ribbons.

The structure has the nonsymmorphic $D_{4h}^{19}$ space group symmetry (No. 141, $I4_1/amd$). Here an important symmetry element is the screw rotation $\bar{C}_{4z}$, which is a fourfold rotation along z followed by a factional lattice translation of $c\hat{z}/4$, where c is the lattice parameter in the z direction. The detailed structural properties are presented in Table 1 and are compared with several other (meta-) stable 3D carbon allotropes. One observes that the cohesive energy of Pentagon Carbon, although slightly higher than graphite and diamond, is less than T6 (ref. 33) and Bcc-C8 (ref. 34) and is comparable with Carbon Kagome Lattice (CKL)[35,36] structure. Due to its porous structure, the mass density of Pentagon Carbon is low ($\sim 2.18\,\mathrm{g\,cm^{-3}}$) comparable with that of graphite. This is also correlated with its relatively low bulk modulus $\sim 257\,\mathrm{GPa}$ compared with other allotropes (except for graphite), suggesting that external strain can be more easily applied. The phonon spectrum of Pentagon Carbon is also investigated, which shows no soft mode, confirming its dynamical stability (more analysis of lattice structure and stability can be found in Supplementary Fig. 1 and Note 1).

**Electronic structure and quantum phase transition.** Remarkably, Pentagon Carbon is also a material hosting a series of novel topological fermions beyond the usual Dirac and Weyl fermions. The band structure in Fig. 2a shows that, in equilibrium state, it is a narrow-gap semiconductor with a small energy gap $\sim 21.4\,\mathrm{meV}$. The electronic structure shows strong anisotropy between in-plane (xy) and out-of-plane (z) directions in accordance with the crystal structural anisotropy. Focusing on the $\Gamma$ point, there are two valence bands (heavy-hole and light-hole) and a single conduction band at low energy, with the conduction band and the light-hole valence band almost crossing linearly. Note that the degeneracy of the two valence bands at $\Gamma$ is dictated by symmetry: the two states belong to a 2D irreducible representation $E_g$ of the $D_{4h}$ group at $\Gamma$. In comparison, the conduction band belongs to the one-dimensional (1D) $A_{1u}$ representation. Another salient feature is that the two valence bands are degenerate along the $\Gamma$–Z path ($k_z$-axis). This is because that there is remaining $C_{4v}$ little group symmetry along this screw axis, such that the 2D representation does not split according to the compatibility relation. From the analysis of wave-function and projected density of states (PDOS), we find that the low-energy states are mainly from the $\pi$ orbitals of the $sp^2$ atoms (Fig. 2f, Supplementary Fig. 2).

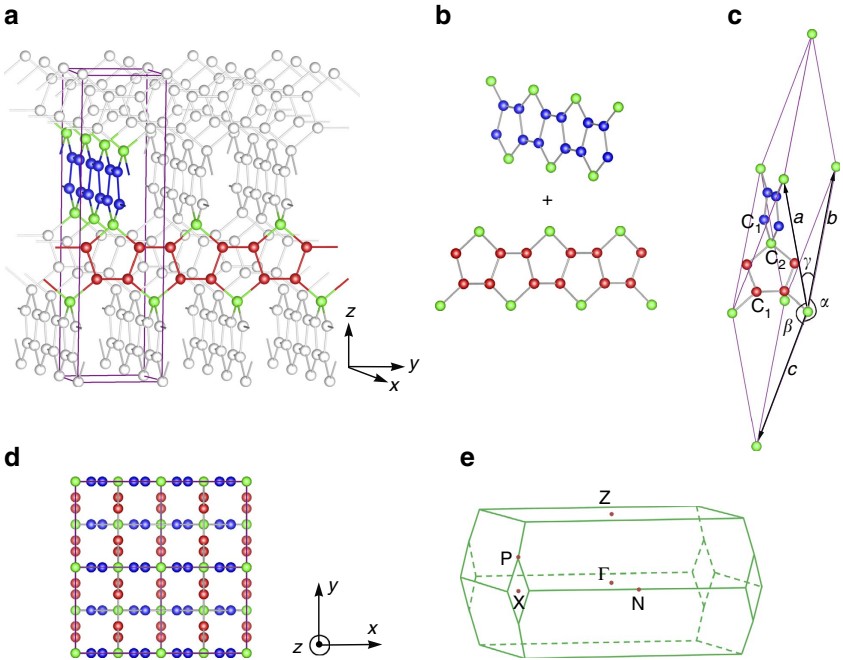

**Figure 1 | Crystal structure and Brillouin zone of 3D Pentagon Carbon.** (**a**) Structure of 3D Pentagon Carbon, where the purple lines depict the conventional unit cell, and the red and blue Carbon atoms form mutually orthogonal armchair chains which are linked by the green atoms. This structure can also be viewed as composed of orthogonal pentagonal rings sharing the vertex green atoms, as shown in **b**. (**c**) Primitive cell of 3D Pentagon Carbon. The red and blue atoms are marked as C1 sites and the green atoms are marked as C2 sites, indicating their different orbital hybridization character. The lattice parameters of the primitive cell are $a = b = c = 7.15$ Å, $\alpha = \beta = 149.86°$, $\gamma = 43.11°$. (**d**) Top view (from z-axis) of the structure, showing that the red and blue armchair chains are mutually orthogonal. (**e**) Brillouin zone of the primitive cell along with the high-symmetry points marked.

Since the semiconducting gap is small, a slight perturbation may close the gap and drive a semiconductor-to-metal phase transition (a tight-binding model analysis is given in Supplementary Figs 3,4 and Note 2). In Fig. 2b–d, we show band structures for three applied uniaxial strains $\varepsilon = 1$, $-0.1$ and $-1\%$ along the $z$ direction. One observes that indeed a semiconductor–metal phase transition is induced. A tensile strain drives the system to a semiconductor with wider gap, whereas a compressive strain drives the system towards a metal. Of particular interest is the change of band ordering at the $\Gamma$ point. The $A_{1u}$ singlet state and the $E_g$ doublet switch their positions in energy across the transition. Due to their different symmetry character, $A_{1u}$ and $E_g$ states cannot hybridize in the transition. Hence at the critical strain, we must have the three bands crossing at a single point, where the conduction and the light-hole bands have linear dispersion in the $k_x$–$k_y$ plane; and all three bands have quadratic dispersion along the screw axis. The 3D band structures in Fig. 2i,j clearly show the anisotropic feature around the triplet point.

Passing the critical point, the band ordering is inverted at $\Gamma$. However, the corresponding states at Z on the Brillouin zone boundary still have the original ordering. This band topology means that the two bands must cross each other between $\Gamma$ and Z. Further, because the two bands still belong to different representations along the screw axis, they must cross in a linear manner with two triply degenerate band-crossing points (one on each side of $\Gamma$), as confirmed in Fig. 2d. Therefore, the metallic phase represents a novel TM phase with a pair of triply degenerate band-crossing points near the Fermi level. In such sense, the quantum phase transition may also be regarded as a topological phase transition from a trivial semiconductor phase to a TM phase.

We can define a $Z_2$ invariant to characterize this TM phase and the phase transition. Let $\Delta$ and $\Delta_z$ be the values of the energy difference between A band and E band at $\Gamma$ point and at Z point,

respectively. Then a $Z_2$ invariant can be defined as $\chi = \mathrm{sgn}(\Delta \cdot \Delta_Z) \in \{+1, -1\}$. $\chi = +1$ corresponds to the trivial case where a band-crossing point does not need to occur, while $\chi = -1$ is the topologically nontrivial case where a triply degenerate crossing point must exist between $\Gamma$ and Z. The topological phase transition from a trivial semiconductor phase to TM phase (as shown in Fig. 2) is thus characterized by $\chi$ changing from $+1$ to $-1$. The topological character is manifest in the sense that as long as the $Z_2$ invariant $\chi$ is well defined (that is, symmetry is preserved) and does not change, the existence of the triply degenerate crossing point and the linear band dispersion along $k_z$ near the point are guaranteed and insensitive to system parameters and perturbations.

**Emergent fermions and $\mathbf{k} \cdot \mathbf{p}$ model.** Nontrivial band-crossing points are fascinating objects because the fermionic quasiparticles around these points necessarily have a multi-component structure, distinct from the usual Schrodinger fermions. The essential physics of these emergent fermions are characterized by the low-energy effective $\mathbf{k} \cdot \mathbf{p}$ model expanded around the crossing point.

Around the $\Gamma$ point, we can construct the $\mathbf{k} \cdot \mathbf{p}$ model based on the three basis states of $A_{1u}$ and $E_g$. Constrained by the $D_{4h}$ symmetry group and the time reversal symmetry for a spinless system (as spin–orbit coupling is negligible for carbon), one obtains the $\mathbf{k} \cdot \mathbf{p}$ model up to $k$-quadratic order as

$$H(\mathbf{k}) = \left(C + D_1 k_z^2 + D_2 k_\perp^2\right) + \begin{bmatrix} \Delta + B_1 k_z^2 + B_2 k_\perp^2 & -iAk_x & -iAk_y \\ iAk_x & 0 & B_3 k_x k_y \\ iAk_y & B_3 k_x k_y & 0 \end{bmatrix},$$

$$(1)$$

where the coefficients $A$, $B_1$, $B_2$, $C$, $D_1$, $D_2$, and $\Delta$ can be determined by fitting the density functional theory (DFT) result. At transition point, $\Delta = 0$, the three bands cross at a single point,

**Table 1 | Comparison between Pentagon Carbon and other representative carbon allotropes.**

| System | Space Group | $d$ ($sp^2$) | $d$ ($sp^3$) | $\theta$ ($sp^2$) | $\theta$ ($sp^3$) | $\rho$ | $B$ | $E_{coh}$ |
|--------|-------------|-----------|-----------|------------|------------|--------|-----|-----------|
| CKL | $F6_3/MMC$ | | 1.53, 1.50 | | 60.0, 115.1, 117.7 | 2.75 | 327 | − 8.81 |
| Bcc C8 | $P4/MMM$ | | 1.52, 1.58, 1.59 | | 90.0, 117.6, 139.1 | 2.97 | 343 | − 8.49 |
| 3D-C5 | $I4_1/amd$ | 1.37, 1.44 | 1.53 | 109.7, 110.3, 140.1 | 100.1, 114.3 | 2.18 | 257 | − 8.79 |
| T6 | $P4_2/MMC$ | 1.34 | 1.54 | 115.1, 122.5 | 106.7, 115.1 | 2.95 | 350 | − 8.63 |
| Diamond | $FD\overline{3}M$ | | 1.54 | | 109.5 | 3.55 | 431 | − 9.09 |
| Graphite | $F6_3/MMC$ | 1.42 | | 120 | | 2.24 | 36 | − 9.22 |

The table contains information of the space group, the bond length $d$ (Å), the bond angle $\theta$ (°), the mass density $\rho$ (g cm$^{-3}$), the bulk modulus $B$ (GPa) and the cohesive energy $E_{coh}$ (eV per atom).

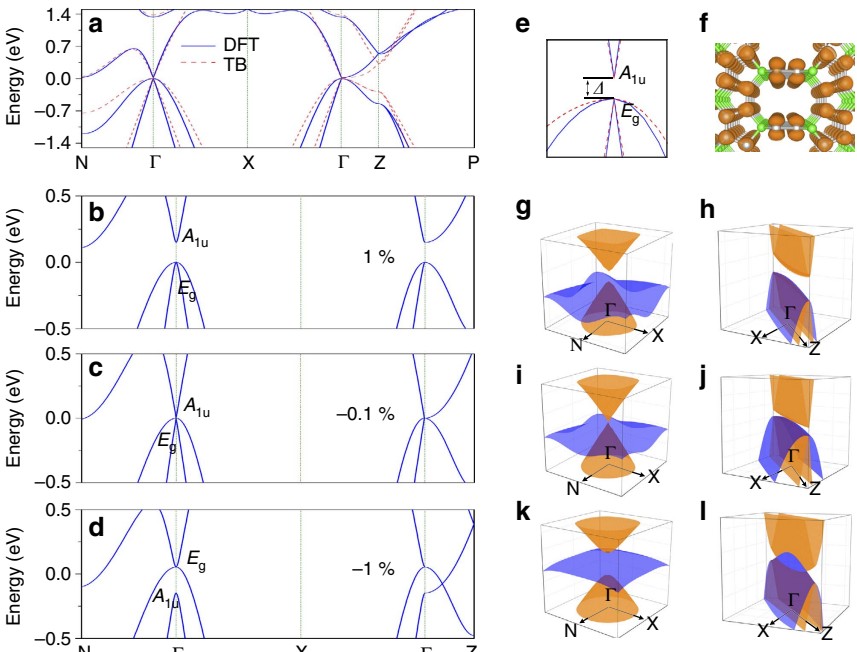

**Figure 2 | Electronic band structure of 3D Pentagon Carbon.** (**a**) Band structure of 3D Pentagon Carbon, as from DFT calculation (blue solid line) and from tight-binding model (red dashed line). The solid and dashed lines are DFT and tight-binding results, respectively. (**b**–**d**) The band structures of 3D Pentagon Carbon under uniaxial strain along z-direction with $\varepsilon = 1\%$, − 0.1% and − 1%, respectively. (**e**) The zoom-in view of band structure around the $\Gamma$ point in **a** along N–$\Gamma$–X. The two degenerate valence bands at $\Gamma$ belong to the two-dimensional irreducible representation $E_g$, and the lowest conduction band belongs to the one-dimensional $A_{1u}$ representation. (**f**) The charge density distribution for states near the Fermi level in **a**, indicating these states are mainly from $\pi$ orbitals of the $sp^2$ hybridized atoms. (**g**–**l**) The 3D band structures on the N–$\Gamma$–X and the X–$\Gamma$–Z planes corresponding to **b**–**d**, respectively.

as illustrated in Fig. 3a. Then we can further simplify the model by keeping only the k-linear terms:

$$\mathcal{H}(\mathbf{k}) = A\mathbf{k} \cdot \boldsymbol{\lambda}, \qquad (2)$$

where $\mathbf{k} = (k_x, k_y, 0)$, and

$$\lambda_x = \begin{bmatrix} 0 & -i & 0 \\ i & 0 & 0 \\ 0 & 0 & 0 \end{bmatrix}, \ \lambda_y = \begin{bmatrix} 0 & 0 & -i \\ 0 & 0 & 0 \\ i & 0 & 0 \end{bmatrix}, \ \lambda_z = \begin{bmatrix} 0 & 0 & 0 \\ 0 & 0 & -i \\ 0 & i & 0 \end{bmatrix}$$

are three of the eight Gell-Mann matrices corresponding to spin-1 (ref. 37). This describes an isospin-1 triplet fermion moving in the xy plane. The fermion is helical, with a well-defined helicity of $\pm 1$ and 0 corresponding to the eigenvalues of the helicity operator $\mathbf{k} \cdot \boldsymbol{\lambda}/k$. The two branches with helicity $\pm 1$ are massless while the remaining one with helicity 0 has a flat dispersion, and these are constrained by the presence of an emergent 'chiral symmetry' operation $\mathcal{C} = \text{diag}(1, -1, -1)$ with $\{H(\mathbf{k}), \mathcal{C}\} = 0$. The finite mass term for $\Delta \neq 0$ breaks this symmetry, generating a mass for both $\pm 1$ branches and gapping the spectrum.

Passing the critical point, the isospin-1 triplet fermion point split into two triply degenerate points located at $k_z = \tau K_c$, as shown in Fig. 3b, where $\tau = \pm$ is an index labelling the two crossing points and $K_c = \sqrt{-\Delta/B_1}$ (note that we must have $\Delta < 0$ and $B_1 > 0$ to reproduce the band structure). The emergent triply degenerate fermions around each crossing point are described by the linearized model obtained from equation (1) as

$$H_\tau(\mathbf{q}) = \begin{bmatrix} 2\tau(B_1 + D_1)K_c q_z & -iAq_x & -iAq_y \\ iAq_x & 2\tau D_1 K_c q_z & 0 \\ iAq_y & 0 & 2\tau D_1 K_c q_z \end{bmatrix}, \quad (3)$$

where the wave vector $\mathbf{q}$ is measured from the crossing point. One notes that in the direction perpendicular to the screw axis, that is, in the $q_x$–$q_y$ plane, the dispersion is the same as that for the isospin-1 triplet fermion in equation (2). One finds that the triply degenerate fermions basically inherit the structure of the triplet fermions but acquire one more degree of freedom, that is, the motion along the z direction. In the meantime, they lose the well-defined helicity.

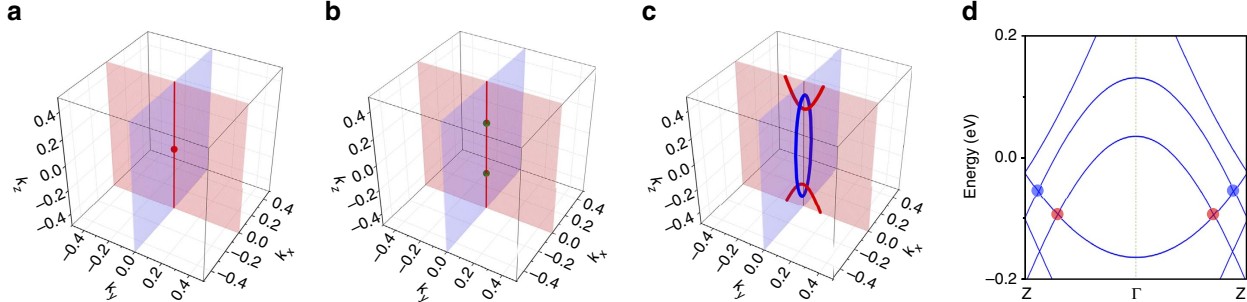

**Figure 3 | Emergent fermions in Pentagon Carbon.** Schematic of the band-crossing evolution in reciprocal space, from (**a**) a single isospin-1 triplet fermion point at the Brillouin zone centre, to (**b**) two triply degenerate fermion points along $k_z$-axis (one on each side of $\Gamma$ point), further to (**c**) two inter-connected (Hopf link) Weyl loops (with one Weyl loop being centered at $\Gamma$ point, while the other loop crossing the first Brillouin zone boundary). In **a,b**, the red line along the $k_z$-axis marks the (twofold) band degeneracy line protected by the fourfold screw-rotational symmetry. Note that the dispersion transverse to this line (that is, in $k_x$–$k_y$ plane) is of quadratic type (which can be observed from Fig. 2). (**d**) The DFT band structure of 3D Pentagon Carbon under broken fourfold screw-rotational symmetry (with uniaxial strain of $-1\%$ along $z$-axis followed by 1% tensile strain along $x$-axis). The red (blue) dot indicates the crossing point of red (blue) circle in **c** with the $k_z$-axis.

The two kinds of fermionic quasiparticles, the isospin-1 triplet fermions and the triply degenerate fermions, are new types of emergent fermions. Unlike the Dirac and Weyl fermions studied before, these new fermions do not have direct analogues in the relativistic quantum field theory. The isospin-1 triplet fermion point is not topologically protected: it marks the critical point in the quantum phase transition. In contrast, the two triply degenerate fermion points are protected by the nontrivial band topology and the fourfold screw axis. These features are schematically illustrated in Fig. 3a,b.

When further breaking the screw-rotational symmetry, the double degeneracy of the E band will be lifted; hence the triply degenerate fermion point will disappear. However, note that the original A and E bands have different eigenvalues under the mirror operations $M_{xz}$ and $M_{yz}$. Hence crossings between each split E band and A band in the mirror-invariant $k_x$–$k_z$ and $k_y$–$k_z$ planes will be protected if the respective mirror symmetry remains. For example, by an additional uniaxial strain along $x$ direction or along diagonal directions in the $x$–$y$ plane, we find that, interestingly, the resulting band crossings form two orthogonal concatenated Weyl loops, which has the topology of a Hopf link. As shown in Fig. 3c, for an additional tensile strain along the $x$ direction, one Weyl loop is centred at the $\Gamma$ point lying in the $k_x$–$k_z$ plane, while the other loop is in the $k_y$–$k_z$ plane and crosses the first Brillouin zone boundary. Around each point on the loop, the energy dispersion along two transverse directions is of the Weyl-like linear type as shown in Fig. 3d. It needs to be emphasized that the band crossing forms the Weyl-loop fermion, not Weyl fermion, because the inversion symmetry is still preserved. It is particularly interesting that the two Weyl loops form an inter-connected Hopf-link pattern, and these two loops are protected by the $M_{xz}$ and $M_{yz}$ mirror symmetries, respectively. It should also be noted that there is no symmetry to pin these Weyl loops at a constant energy, and, indeed, the energy varies along each loop.

**Landau level spectrum.** Topological character of the band structure can manifest when subjecting to a magnetic field[38–40]. For a 3D system, the electron motion in the plane perpendicular to the field is quantized into LLs. The dispersion along the field connects the levels to form dispersive Landau bands. For example, at each Weyl point, there is a gapless chiral Landau band, such that an additional parallel electric field will drive charge creation or depletion depending on the chirality. Since Weyl points appear in pairs of opposite chirality, the process

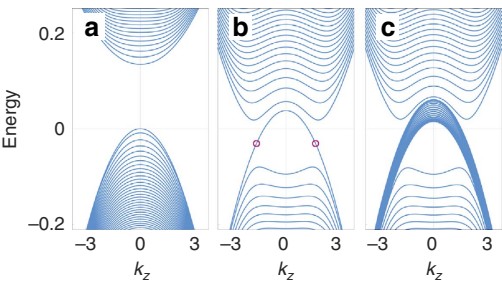

**Figure 4 | Landau level spectra.** Landau level spectra for (**a**) the semiconductor phase, (**b**) the TM phase (corresponding to Fig. 2d) and (**c**) the TM phase with further screw-rotational symmetry breaking (corresponding to Fig. 3c). The magnetic field is along the $z$-direction. The red circles in **b** indicate the location of the two triply degenerate fermion points (Fig. 3b).

leads to charge pumping between the Weyl points. This non-conservation of chiral charge at a single Weyl point is the well-known chiral anomaly[14,18,40]. Experimentally, it manifests as a negative magnetoresistance, which has recently been demonstrated in Weyl semimetals[15,16].

For Pentagon Carbon, considering a magnetic field along the screw axis such that the screw rotational symmetry is preserved. Focusing on the low-energy physics around the $\Gamma$ point where the quantum phase transition occurs, we find that the LL spectrum exhibits drastic difference across the transition. As illustrated in Fig. 4a, one observes that in the semiconductor phase, the LL spectrum shows a spectral gap, typical for a semiconducting state. Remarkably, in the TM phase, there appear two chiral Landau bands crossing the spectral gap and crossing the location of the original triply degenerate fermion points (Fig. 4b). This is like that of a Weyl point; however, there is a marked difference: the chiral band here is in fact with a huge number of Landau bands nearly degenerate onto a single curve[21]. This behaviour is due to the small values of the in-plane quadratic terms such as $B_2$ and $B_3$. If one artificially increases these parameters, there will be a visible splitting between the almost degenerate Landau bands. Similar to chiral anomaly, a parallel **E**-field would pump charges from one triply degenerate point to the other. Due to the large number of the bands, it is expected to generate pronounced signals in magneto-transport experiment.

When further breaking the screw-rotational symmetry, the triply degenerate fermion points disappear, giving rise to inter-connected Hopf-link Weyl loops. In the LL spectrum, we see that the main effect of the symmetry breaking is to split the chiral Landau bands, as indicated in Fig. 4c. These interesting LL features reflect the evolution of the band-crossing points and will be useful for probing the quantum phase transition and the emergent topological fermions.

## Discussion

Our work discovers a new metastable 3D carbon allotrope—the Pentagon Carbon, which exhibits a beautiful lattice structure entirely composed of pentagonal rings—a highly rare lattice motif for 3D crystals. Originating from this unique structure, there appear interesting quantum phase transitions and the associated novel emergent fermions. Remarkably, we reveal a fascinating transformation from isospin-1 triplet fermion point to triply degenerate fermion point and finally to inter-connected Hopf-link Weyl loops.

Nontrivial band-crossing point is also predicted in a few recent works. Particularly, Bradlyn *et al.*[21] classified all possible degeneracy points at high-symmetry $k$-points for the 230 space groups in spin–orbit-coupled systems, and triple-degenerate points have also been found in a few materials with strong spin–orbit coupling[22,41,42]. The triply degenerate points here are distinct from those previous examples in the following aspects: first, due to the negligible SOC, Pentagon Carbon is essentially a spinless system, and, as we mentioned at the beginning, spinful and spinless systems belong to different symmetry/topological classes (For example, the time reversal operation has distinct properties between the two cases. And rotations for spinful systems will also operate on the spin degree of freedom, requiring a double group representation, distinct from spinless systems.); second, while the degeneracy points in ref. 21 are at high-symmetry points, here they are on symmetry lines ($\Gamma$–$Z$). Hence, without altering the symmetry, the latter is free to move along the symmetry line by applying a strain with observable physical consequences, whereas the former is pinned at the high-symmetry $k$-points. A work by Heikkila and Volovik discussed possible triply degenerate points in Bernal graphite[27], where each point connects four Dirac band-crossing lines with each line protected by a $Z_2$ invariant $N_1 = \pm 1$. In contrast, the triply degenerate points in Pentagon Carbon are on a single-band degeneracy line along $k_z$-axis (as in Fig. 3b); it is not a Dirac line because the dispersion is quadratic along the transverse directions, so the corresponding $Z_2$ invariant, as defined in ref. 27, is trivial ($N_1 = 0$). The protection of the band degeneracy line and triply degenerate point here has a qualitatively different physical origin, which is originated from the fourfold screw-rotational symmetry and band ordering topology, not present in Bernal graphite. We note that the different configurations of degeneracy lines in Bernal graphite and Pentagon Carbon precisely reflect their distinct symmetries: the arrangement of four Dirac lines with $Z_2$ indices ($-1, +1, +1, +1$) observed in Bernal graphite is allowed by its threefold rotational symmetry, but it is not compatible with the fourfold screw-rotation in Pentagon Carbon.

Here, we focus on the effect of uniaxial strain in driving the phase transition. Other types of lattice deformation can also produce a similar effect, as long as the fourfold screw-rotational symmetry is preserved. For example, isotropic lattice strain or hydrostatic pressure can drive the phase transition with qualitatively similar features: the critical value is $-1\%$ for isotropic strain and $4.75$ GPa for hydrostatic pressure. We also mention that like other carbon allotropes, Pentagon Carbon has

excellent mechanical property. From first-principles calculation, it shows a linear elastic regime up to $\pm 10\%$ strain (Supplementary Fig. 5). Hence the strain-induced phase transitions discussed here should be readily achievable.

To guide experiment, we simulate X-ray diffraction peaks of Pentagon Carbon and contrast them with those of representative carbon allotropes (Supplementary Fig. 6). We also suggest a possible reaction route for the chemical synthesis. Molecules with pentagonal rings can be synthesized, for example, in cyclopentadiene with different substitutes. Starting from 1,4-dibromocyclopentadiene, the cyclopentadiene building units can be linked into a 1D ribbon through Yamamoto polymerization. The 1D ribbons could be functionalized with bromides and boronic acids which are connected into the 3D framework through Suzuki coupling (the details are described in Supplementary Fig. 7). Each step in this process is either already achieved or have a high chance to be achieved because of the existence of similar reactions. Given its energetic and dynamic stabilities and in view of the rapid progress in experimental techniques which have realized a number of carbon structures in recent decade[34,43], we expect that Pentagon Carbon could also be realized in the near future.

## Methods

**Computational methods.** We performed first-principles calculations within the DFT formalism as implemented in VASP code[44,45]. The potential of the core electrons and the exchange-correlation interaction between the valence electrons are described, respectively, by the projector augmented wave[46] approach and the generalized gradient approximation in the Perdew–Burke–Ernzerhof[47] implementation. The kinetic energy cutoff of 500 eV is employed. The atomic positions were optimized using the conjugate gradient method. The energy and force convergence criteria are set to be $10^{-5}$ eV and $10^{-2}$ eV Å$^{-1}$, respectively. The band structure and projected density of states (PDOS) were calculated using the primitive cell. The Brillouin zone was sampled with a $7 \times 7 \times 7$ Monkhorst–Pack special $k$-point grid for geometrical optimization. Phonon spectra are calculated using force-constants method, and the dynamical matrices are computed using the finite differences method in a $2 \times 2 \times 2$ supercell to eliminate errors in the low frequency modes (calculation with $3 \times 3 \times 3$ supercell is also carried out for verification). The force constants and phonon spectra were obtained using the Phonopy package[48]. Calculation with the Heyd–Scuseria–Ernzerhof (HSE06) hybrid functional[49] is also performed to verify the band structure. The values of the band gap and the critical transition strain become larger compared with Perdew–Burke–Ernzerhof results (the HSE06 gap is about 471 meV and the critical strain is about $-2\%$), but the qualitative features of the bands and phase transitions remain the same (Supplementary Fig. 8 and Note 3). The effect of van der Waals interaction is also tested and is found to be weak, for example, by including the van der Waals interaction, the cohesive energy is decreased from $-8.79$ to $-8.89$ eV and the lattice constants are decreased by about 0.2% (Supplementary Fig. 9).

**Data availability.** The data that support the findings of this study are available from the corresponding authors upon reasonable request.

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

## Acknowledgements

We thank the helpful discussions with Man-Fung Cheung, D.L. Deng, Shuai Yuan, Weiyu Xie and Damien J. West. This work was supported by the National Natural Science Foundation of China (Nos. 51376005 and 11474243). S.Z. acknowledges the support by US DOE under Grant No. DE-SC0002623. Z.-M.Y. and S.A.Y. are supported by the Singapore MOE Academic Research Fund Tier 1 (SUTD-T1-2015004) and Tier 2 (MOE2015-T2-2-144).

## Author contributions

Y.C. proposed the Pentagon Carbon. Y.C., S.A.Y. and S.Z. conceived the original ideas. C.Z., Y.C., Z.-M.Y. and S.A.Y. conducted the calculations. Y.C. and S.A.Y. wrote the manuscript. All authors discussed the results and commented on the manuscript at all stages.

## Additional information

**Competing interests:** The authors declare no competing financial interests.

