## [Peer Review File · Nature Communications]

Reviewers' comments:

Reviewer #1 (Remarks to the Author):

The authors propose a new three-dimensional form of carbon crystal based on pentagons as the elementary motif. Based on first principle calculations, they show that this pentagon carbon is stable and its electronic structure host a number of interesting band degeneracies that make it a topological semimetal. In contrast to many topological semimetals discussed recently, this compound has negligible spin-orbit coupling, so that the electronic structure can be thought of as being spinless.

I think that the subject of the manuscript is timely and interesting, and the material is presented in a form that is overall appropriate and sound.

The most critical point about the paper, which is hard to judge on, is how likely pentagon carbon can actually be synthesized. The authors discuss a possible reaction path.

Aside from this fundamental question, my main criticism is that the manuscript misses some relevant references in the field of topological semimetals. In particular for a manuscript concerned with spinless fermions, <https://arxiv.org/abs/1505.03277> should be cited and its relation to the topological band degeneracies of the pentagon carbon discussed. In fact, this paper discusses for usual carbon a band structure that — as far as I understand — is qualitatively the same as the one discussed here for pentagon carbon. (It also hosts the triply degenerate points.) Also, Bradlyn et al., *Science*, 353, aaf5037 (2016) should be referenced, as it discusses a similar pseudospin-1 Hamiltonian. In view of these papers, it is also not clear to me why the authors call the triply degenerate band touching point 'novel'. Very similar band degeneracies have been discussed in the above. The same is true for the Landau level spectra.

Also, I am confused about the Landau level spectrum in Fig 4b. If the middle band is to be fully degenerate, it should be absolutely non-dispersing in the direction perpendicular to the magnetic field direction. However, I do not see which symmetry would guarantee such an absolutely flat band and neither do I see a completely flat band in Fig. 2d.

Finally, the language can be improved in several places.

In summary, I think that the paper is not publishable in *Nature Communications* in this form, as the exotic electronic properties discussed for pentagon carbon seem to be closely related to those found in <https://arxiv.org/abs/1505.03277>. If the authors explain what specifically is novel about the electronic structure that they discuss, as compared to previous studies, the manuscript could be reconsidered.

Reviewer #2 (Remarks to the Author):

In this paper, the authors proposed a new form of carbon: pentagon carbon entirely composed of pentagon rings. The sp^2 -hybridized carbon pentagon motifs are interlinked by sp^3 -hybridized carbon atoms. The pentagon carbon exhibits successive topological phase transitions under isotropic strain followed by uniaxial strain, generating novel quasiparticles, such as isospin-1 triplet fermions, triply-degenerate fermions, and orthogonal concatenated Weyl loops. The results are very interesting, providing the unique electronic structure of pentagon carbon. However, there is a lack of some convincing evidences for their arguments. Thus, the manuscript in the present form is not appropriate for publication in *Nature Communications*. The details of comments are listed below.

1. On page 4, the authors argued that the topological metallic phase is obtained. However, its

evidence was not clearly given, while the band ordering and the symmetry of wave functions were discussed. In general, the band ordering change does not guarantee the topological phase transition. To clearly demonstrate the topological phase transition, the topological invariant needs to be examined, such as Z2 invariant in topological insulators.

2. In the proposed carbon allotrope, the pentagon motifs consist of sp² bonded atoms. In sp²-hybridized carbon systems, the atomic structure and total energy are very sensitive to the type of the exchange-correlation functional and van der Waals forces. For instance, the volume of graphite is not well described by the PBE functional without corrections. In this work, all the calculations were performed by using the PBE exchange-correlation functional. The authors need to discuss the effect of van der Waals forces.

3. Based on the PBE band structure, it was concluded that the pentagon carbon is semimetallic (Fig. 2a). However, in Figure 2(a), it seems that the conduction band minimum at the N point is above 0 eV, indicating that it may be a semiconductor. The authors should make this point clear. In addition, in Figure 2(a), the band gap at Γ is extremely small. The small band gap at Γ may be caused by the well-known bandgap underestimation with the PBE functional. The band structure should be checked by improving the band gap, performing advanced DFT calculations, such as the hybrid functional for the exchange-correlation potential. If the band gap at Γ is underestimated, this underestimation may affect the critical strain and the topological phase transition.

4. The electronic structure of the pentagon carbon was investigated under isotropic strain. Since the crystal structure of the pentagon carbon is anisotropic, it may be very difficult to achieve isotropic strain, compared with cubic structures. They have to discuss how to generate isotropic strain in this anisotropic allotrope. Instead of isotropic strain, it may be easier to apply hydrostatic pressure. Then, how do the electronic structure and crystal structure of the pentagon carbon change under hydrostatic pressure?

5. For the calculations of phonon spectrum, they used the 2x2x2 supercell. Because the lattice parameters were not given for the pentagon carbon, it is not clear whether the 2x2x2 supercell is sufficiently large or not. They should provide the lattice parameters for the primitive cell.

6. On page 6, they argued that "the resulting band crossings form two orthogonal concatenated Weyl loops" without clear physical explanations. In what specific way does the uniaxial strain transform the triply-degenerate fermions into the Weyl-loop fermion, instead of several Weyl points. What symmetry protects the Weyl loops?

7. In the manuscript, Fig. 3(a), (b) and (d) were not referred.

8. In the $k \cdot p$ model, they discussed the dependence of topological transition on the parameter Δ . In case of $\Delta=0$, with neglecting the quadratic order in k , the Hamiltonian is simplified with only the linear term. This Hamiltonian describes the isospin-1 triplet fermions with the helicities of ± 1 and 0. However, this triple degeneracy under the strain of -1% is the accidental degeneracy, which can be easily lifted by even weak perturbation. They need to mention explicitly in the manuscript that the isospin-1 triplet point is an accidental point, which is not protected by symmetry, while statements "the isospin-1 triplet fermion point is not topologically protected" are given in Supplementary Information.

9. In many places, words "stable carbon allotrope" are used. In fact, the pentagon carbon is not a stable form of carbon, but a metastable phase with the very high energy of about 0.3 eV relative diamond. Words should be replaced by "metastable carbon allotrope".

Reviewer #1 (Remarks to the Author):

1. *“The authors propose a new three-dimensional form of carbon crystal based on pentagons as the elementary motif. Based on first principle calculations, they show that this pentagon carbon is stable and its electronic structure host a number of interesting band degeneracies that make it a topological semimetal. In contrast to many topological semimetals discussed recently, this compound has negligible spin-orbit coupling, so that the electronic structure can be thought of as begin spinless.*

I think that the subject of the manuscript is timely and interesting, and the material is presented in a form that is overall appropriate and sound.”

Response: We thank the reviewer for considering our work *“timely and interesting”* and *“presented in a form that is overall appropriate and sound”*.

2. *“The most critical point about the paper, which is hard to judge on, is how likely Pentagon Carbon can actually be synthesized. The authors discuss a possible reaction path.”*

Response: We understand the concern how likely a predicted material can be synthesized. However, we do think recent experimental progresses make it more promising for a theoretical proposal to be verified, provided that it is based on a careful examination of the thermodynamic as well as kinetic principles. Taking Bcc-C8 as an example, it was proposed by Johnston and Hoffmann in 1989 (J. Am. Chem. Soc. 111, 810-819, 1989) and was experimentally realized in 2008 with a pulsed-laser induced liquid-solid interface interaction technique (Crystal Growth & Design 8, 581-586, 2008). Although there is a 19-year delay in experimental realization, the power of DFT for carbon systems has been demonstrated nearly three decades ago. With these many years further development, the first-principles DFT methods have become more mature and accurate enough to determine crystal structures and stabilities of carbon. The recent examples are graphyne and boron sheets with much shorter lapses of time before their experimental realization.

On top of this, Pentagon Carbon has good thermodynamic stability with its energy lower than the experimentally realized Bcc-C8 carbon. Moreover, in our discussion of the possible reaction path to Pentagon Carbon, each step is either already achieved or have a high chance to be achieved because

of the existence of similar reactions. We believe following such a design principle could accelerate the final experimental realization of the Pentagon Carbon. To further guide experiment for the identification of Pentagon Carbon, we also calculate X-ray diffraction (XRD) peaks in contrast to other carbon structures.

We have added the XRD results to Supporting Information and the following discussion to the last paragraph in the Discussion Section:

“To guide experiment, we simulate X-ray diffraction (XRD) peaks of Pentagon Carbon and contrast them with those of representative carbon allotropes (see Fig. S7 in the SI). Each step in this process is either already achieved or have a high chance to be achieved because of the existence of similar reactions. Given its energetic and dynamic stabilities and in view of the rapid progress in experimental techniques which have realized a number of carbon structures in recent decade^{38, 45}, we expect Pentagon Carbon could also be realized in the near future.”

3. *“Aside from this fundamental question, my main criticism is that the manuscript misses some relevant references in the field of topological semimetals. In particular for a manuscript concerned with spinless fermions, <https://arxiv.org/abs/1505.03277> should be cited and its relation to the topological band degeneracies of the Pentagon Carbon discussed. In fact, this paper discusses for usual carbon a band structure that — as far as I understand — is qualitatively the same as the one discussed here for Pentagon Carbon. (It also hosts the triply degenerate points.) Also, Bradlyn et al., *Science*, 353, aaf5037 (2016) should be referenced, as it discusses a similar pseudospin-1 Hamiltonian. In view of these papers, it is also not clear to me why the authors call the triply degenerate band touching point ‘novel’. Very similar band degeneracies have been discussed in the above. The same is true for the Landau level spectra.”*

Response: We thank the reviewer and have added the mentioned references in the revised version. Indeed, triply-degenerate point has been discussed in both papers, but they are distinctly different from what we have for the following reasons:

The work by Heikkila and Volovik [<https://arxiv.org/abs/1505.03277>, and New. J. Phys. 17, 093019 (2015)] discussed the band structure of Bernal graphite, in which the crossing points form 4 Dirac lines. Each Dirac line has a linear dispersion along the two transverse directions, and is characterized by a Z_2 topological invariant $N_1=\pm 1$ (see Figure 2, lower panel in that paper). The triply-degenerate point, termed the “nexus point”, corresponds to the merging point of the 4 Dirac lines. In contrast, the band crossing points of Pentagon Carbon form a *single* line along the k_z -axis [see Figure 3(b)], which is *not* a Dirac line because, along the transverse directions, the dispersion is quadratic so the corresponding Z_2 is trivial with $N_1=0$ (as defined in the framework of Heikkila and Volovik). Thus, the band crossing line here is *not* protected by Z_2 invariant, instead, it is protected by a 4-fold screw-rotational symmetry along the k_z -axis. As such, the two bands along the band crossing line belong to the 2D irreducible representation E of the symmetry group. Consequently, the triply-degenerate point here is *not* a “nexus point” that connects multiple Dirac lines, instead, it is formed by the crossing of a singlet A band with the doubly degenerate E band. Thus, it is the specific crystal symmetry in our case that enforces the crossing at the triply-degenerate point without hybridization. In other words, our triply-degenerate point and the one by Heikkila and Volovik represent qualitatively different physics.

On the other hand, the work by Bradlyn *et al.* is a systematic study that searched out band degeneracy points for 230 space groups. Their triply-degenerate points are distinct from ours in at least two fundamental aspects: 1) their classification is explicitly with spin-orbit-coupling (SOC), hence all candidate materials such as $\text{Ag}_3\text{Se}_2\text{Au}$ are heavy for strong SOC; in contrast, due to the negligible SOC, Pentagon Carbon here is spinless. Spinful and spinless systems are not in the same symmetry/topological classes. 2) As a fallout of the 1) above, the degeneracy points by Bradlyn *et al.* are at the high-symmetry points of the Brillouin zone, whereas ours are NOT. They are along the symmetry lines (i.e., the Γ -Z lines), instead. To preserve the symmetry, the degeneracy points of Bradlyn *et al.* are pinned at the high-symmetry points, but our triply-degenerate points are free to move along the symmetry lines, as we have demonstrated by applying a strain.

In summary, the triply-degenerate points in Pentagon Carbon are a new class of triply-degenerate points both in mathematics and in physics, which does not belong to any of the previous classifications, and have exotic electronic properties.

As for the Landau level spectra, because the bands near the triply degenerate points share the similar pseudospin-1 Hamiltonian, our spectra indeed appear similar to those by Bradlyn *et al.* However, the different symmetry classification between the two, as pointed out above, allows us to study the evolution of the Landau level spectra: when the band structure undergoes phase transitions driven by strain, pronounced signals in magneto transport will result. This important feature, however, does not exist in Bradlyn *et al.* model.

Besides the references (Refs [31] and [24] in the revised manuscript), we also added the following discussion on the distinction between our findings and previous works in the Discussion Section:

“Nontrivial band-crossing point is also predicted in a few recent works. Particularly, Bradlyn et al. classified all possible degeneracy points at high-symmetry k-points for the 230 space groups in spin-orbit-coupled systems. The triply-degenerate points here are distinct from that in Ref. 24 in two fundamental aspects: 1) due to the negligible SOC, Pentagon Carbon is essentially a spinless system, and as we mentioned at the beginning, spinful and spinless systems belong to different symmetry/topological classes; 2) Subsequently, while the degeneracy points in Ref. 24 are at high-symmetry points, here they are on symmetry lines (Γ -Z). Thus, without altering the symmetry, the latter is free to move along the symmetry line by applying a strain with observable physical consequences, whereas the former is pinned at the high-symmetry k-points. Heikkila and Volovik found triply-degenerate points in Bernal graphite³¹, where each point connects four Dirac band-crossing lines with each line protected by a Z_2 invariant $N_1=\pm 1$. In contrast, the triply-degenerate points in Pentagon Carbon are on a single band degeneracy line along k_z -axis (as in Figure 3(b)); it is not a Dirac line because the dispersion is quadratic along the transverse directions, so the corresponding Z_2 invariant, as defined in Ref. 31, is trivial ($N_1=0$). The protection of the band degeneracy line and triply-degenerate point here has a qualitatively different physical origin, which is originated from the four-fold screw-rotational symmetry and band ordering topology, not present in Bernal graphite.”

4. *“Also, I am confused about the Landau level spectrum in Fig 4b. If the middle band is to be fully degenerate, it should be absolutely non-dispersing in the direction perpendicular to the magnetic field direction. However, I do not see which symmetry would guarantee such an absolutely flat band and neither do I see a completely flat band in Fig. 2d.”*

Response: The reviewer is correct. The middle band in Fig 4b is not fully degenerate, just nearly degenerate. The fully degenerate case only happens when the band is non-dispersing in directions perpendicular to the magnetic field. The “nearly degenerate” behavior observed here is due to the relatively small dispersion in xy -plane near the Γ point (i.e. due to small values of the coefficients such as B_2 and B_3 in Eq.(1)). Indeed, if one artificially increases B_2 and B_3 , there will be visible splitting of the bands. In the revised manuscript, we clarify this issue by adding the following sentence on Page 12:

“This behavior is due to the small values of the in-plane quadratic terms such as B_2 and B_3 . If one artificially increases these parameters, there will be a visible splitting between the almost degenerate Landau bands.”

5. *“Finally, the language can be improved in several places.”*

Response: We thank the reviewer for the kindly suggestion. We have carefully checked the manuscript to improve the language.

6. *“In summary, I think that the paper is not publishable in Nature Communications in this form, as the exotic electronic properties discussed for Pentagon Carbon seem to be closely related to those found in <https://arxiv.org/abs/1505.03277>. If the authors explain what specifically is novel about the electronic structure that they discuss, as compared to previous studies, the manuscript could be reconsidered.”*

Response: We have explained the distinction between our work and previous works and have revised the manuscript accordingly. We sincerely hope the reviewer can be convinced by the uniqueness of the physics of Pentagon Carbon and recommend the publication of our work.

Reviewer #2 (Remarks to the Author):

1. *“In this paper, the authors proposed a new form of carbon: pentagon carbon entirely composed of pentagon rings. The sp^2 -hybridized carbon pentagon motifs are interlinked by sp^3 -hybridized carbon atoms. The pentagon carbon exhibits successive topological phase transitions under isotropic strain followed by uniaxial strain, generating novel quasiparticles, such as isospin-1 triplet fermions, triply-degenerate fermions, and orthogonal concatenated Weyl loops. The results are very interesting, providing the unique electronic structure of pentagon carbon.”*

Response: We thank the review for his/her comments, especially for pointing out that *“The results are very interesting, providing the unique electronic structure of pentagon carbon.”*

2. *“On page 4, the authors argued that the topological metallic phase is obtained. However, its evidence was not clearly given, while the band ordering and the symmetry of wave functions were discussed. In general, the band ordering change does not guarantee the topological phase transition. To clearly demonstrate the topological phase transition, the topological invariant needs to be examined, such as Z_2 invariant in topological insulators.”*

Response: We thank the referee for the comment. To address it, on Pages 7 and 8 of the revised manuscript, we added the following discussion:

“We can define a Z_2 invariant to characterize this TM phase and the phase transition. Let Δ and Δ_Z be the values of the energy difference between A band and E band at Γ point and at Z point, respectively. Then a Z_2 invariant can be defined as $\chi = \text{sgn}(\Delta \cdot \Delta_Z) \in \{+1, -1\}$. $\chi = +1$ corresponds to the trivial case where a band crossing point does not need to occur, while $\chi = -1$ is the topologically nontrivial case where a triply-degenerate crossing point must exist between Γ and Z. The topological phase transition from a trivial semiconductor phase to TM phase (as shown in Fig.2) is thus characterized by χ changing from +1 to -1. The topological character is manifest in the sense that as long as the Z_2 invariant χ is well defined (i.e. symmetry is preserved) and does not change, the existence of the triply-degenerate crossing point and the linear band dispersion along k_z near the point are guaranteed and insensitive to system parameters and perturbations.”

3. *“In the proposed carbon allotrope, the pentagon motifs consist of sp^2 bonded atoms. In sp^2 -hybridized carbon systems, the atomic structure and total energy are very sensitive to the type of the exchange-correlation functional and van der Waals forces. For instance, the volume of graphite is not well described by the PBE functional without corrections. In this work, all the calculations were performed by using the PBE exchange-correlation functional. The authors need to discuss the effect of van der Waals forces.”*

Response: While van der Waals forces are important for layered structures such as graphite, Pentagon Carbon is a 3D structure for which the effect of van der Waals forces is expected to be smaller. To quantify, we calculated Pentagon Carbon with van der Waals forces included. The results justifies the omission of van der Waals interactions in our previous calculations. For example, the cohesive energy is decreased by 0.10 eV or 1.1%, whereas the lattice constants are decreased by an even less 0.2%. In the revised manuscript, we have added a discussion in the Method Section and a more detailed discussion in the Supporting Information to address this issue.

4. *“Based on the PBE band structure, it was concluded that the Pentagon Carbon is semimetallic (Fig. 2a). However, in Figure 2(a), it seems that the conduction band minimum at the N point is above 0 eV, indicating that it may be a semiconductor. The authors should make this point clear. In addition, in Figure 2(a), the band gap at Γ is extremely small. The small band gap at Γ may be caused by the well-known bandgap underestimation with the PBE functional. The band structure should be checked by improving the band gap, performing advanced DFT calculations, such as the hybrid functional for the exchange-correlation potential. If the band gap at Γ is underestimated, this underestimation may affect the critical strain and the topological phase transition.”*

Response: The structure without strain is indeed a semiconductor. In the revised manuscript, we have changed the wording to make this point clear.

Following the suggestion of the reviewer, we calculated the band structures by using hybrid functional HSE06, obtaining a band gap of 0.471 eV. As mentioned by the reviewer, this affects the

quantitative value of the critical strain: Figure R1 below shows that the phase transition now takes place at $\sim -2\%$ uniaxial strain. It is known that Hartree-Fock method fails in metal as it yields a zero density of states at the Fermi level. This drawback is expected to affect the hybrid methods, so the 2% strain here should be an upper bound. The important point is that the qualitative features of the band structure and topological phase transition remain unchanged.

We have added the discussion to the Method Section in the revised manuscript, with more details given in the Supporting Information.

Figure R1 | A comparison of the band structures between PBE and HSE06 for Pentagon Carbon at different uniaxial strain (marked).

5. “The electronic structure of the Pentagon Carbon was investigated under isotropic strain. Since the crystal structure of the Pentagon Carbon is anisotropic, it may be very difficult to achieve isotropic strain, compared with cubic structures. They have to discuss how to generate isotropic strain in this anisotropic allotrope. Instead of isotropic strain, it may be easier to apply hydrostatic pressure. Then, how do the electronic structure and crystal structure of the Pentagon Carbon change under hydrostatic pressure?”

Response: Indeed, isotropic strain could be challenging to realize. In comparison, hydrostatic pressure or uniaxial strain (along z-direction) are much easier. We found that, with a hydrostatic

pressure, the topological phase transition happens at 4.75 GPa (see Figure R2 below), while with a uniaxial strain, it happens at -0.1% on the PBE level and -2% on the HSE06 level, respectively. Note that all the main features in Figure R2 remain the same. Therefore, in the revised manuscript, we only focus on the result of uniaxial strain. These new results are discussed in the Discussion Section:

“Here we focus on the effect of uniaxial strain in driving the phase transition. Other types of lattice deformation can also produce a similar effect, as long as the four-fold screw-rotational symmetry is preserved. For example, isotropic lattice strain or hydrostatic pressure can drive the phase transition with qualitatively similar features: the critical value is -1% for isotropic strain and 4.75 GPa for hydrostatic pressure.”

Figure R2 | A comparison of the PBE band structures with (a) -0.1 % uniaxial strain, (b) -1% for isotropic strain, and (c) 4.75 GPa hydrostatic pressure.

6. *“For the calculations of phonon spectrum, they used the 2x2x2 supercell. Because the lattice parameters were not given for the Pentagon Carbon, it is not clear whether the 2x2x2 supercell is sufficiently large or not. They should provide the lattice parameters for the primitive cell.”*

Response: Lattice parameters for the primitive cell have been provided in the revised caption of Figure 1 to read *“The lattice parameters of the primitive cell are $a = b = c = 7.15 \text{ \AA}$, $\alpha = \beta = 149.86^\circ$, $\gamma = 43.11^\circ$ ”*. With such a size, the 2x2x2 supercell should be sufficient for phonon calculation. To confirm there is no imaginary frequency, we also calculated force constants using a 3x3x3 supercell. The result is shown in Figure R3 (below), which is nearly the same to that of 2x2x2 supercell

Figure R3 | Phonon spectrum of Pentagon Carbon, with force constants obtained from a 3x3x3 supercell.

7. “On page 6, they argued that “the resulting band crossings form two orthogonal concatenated Weyl loops” without clear physical explanations. In what specific way does the uniaxial strain transform the triply-degenerate fermions into the Weyl-loop fermion, instead of several Weyl points. What symmetry protects the Weyl loops?”

Response: A uniaxial strain along x (or y) direction breaks the 4-fold screw-rotational symmetry and lifts the double-degeneracy of the E band, so the triply-degenerate points will disappear. Such a distortion, however, does not break inversion symmetry, so isolated Weyl points cannot appear. Only Weyl loops can appear. The two concatenated loops in the paper are protected by mirror symmetries: The original E and A bands have different eigenvalues under mirror operations, M_{xz} and M_{yz} . Hence, crossings between each split E band and A band in the mirror-invariant k_x - k_z and k_y - k_z planes will be protected if the respective mirror symmetry remains. In the revised manuscript, we add the following detailed discussion on this point on Pages 10 and 11:

“As shown in Fig. 3(c), for an additional tensile strain along the x -direction, one Weyl loop is centered at the Γ point lying in the k_x - k_z plane, while the other loop is in the k_y - k_z plane and crosses the first Brillouin zone boundary. Around each point on the loop, the energy dispersion along two transverse directions is of the Weyl-like linear type, as shown in Fig. 3(d). It needs to be emphasized that the band crossing forms the Weyl loop fermion, not Weyl fermion, because the inversion symmetry is still preserved. It is particularly interesting that the two Weyl loops form an

inter-connected pattern. And these two loops are protected by the M_{xz} and M_{yz} mirror symmetries respectively. It should also be noted that there is no symmetry to pin these Weyl loops at a constant energy, and indeed, the energy varies along each loop.”

8. *“In the manuscript, Fig. 3(a), (b) and (d) were not referred.”*

Response: We thank the reviewer for pointing out this omission. We have added description of these figures in the revised manuscript.

9. *“In the $k \cdot p$ model, they discussed the dependence of topological transition on the parameter Δ . In case of $\Delta=0$, with neglecting the quadratic order in k , the Hamiltonian is simplified with only the linear term. This Hamiltonian describes the isospin-1 triplet fermions with the helicities of ± 1 and 0. However, this triple degeneracy under the strain of -1% is the accidental degeneracy, which can be easily lifted by even weak perturbation. They need to mention explicitly in the manuscript that the isospin-1 triplet point is an accidental point, which is not protected by symmetry, while statements “the isospin-1 triplet fermion point is not topologically protected” are given in Supplementary Information.”*

Response: In revised manuscript, we brought this point to page 10 of the main text which reads:

“The isospin-1 triplet fermion point is not topologically protected: it marks the critical point in the quantum phase transition. In contrast, the two triply-degenerate fermion points are protected by the nontrivial band topology and the four-fold screw axis. These features are schematically illustrated in Figs. 3(a- b).”

10. *In many places, words “stable carbon allotrope” are used. In fact, the Pentagon Carbon is not a stable form of carbon, but a metastable phase with the very high energy of about 0.3 eV relative diamond. Words should be replaced by “metastable carbon allotrope”.*

Response: We agree and have replaced the words “stable carbon allotrope” by “metastable carbon allotrope” in the revised manuscript.

REVIEWERS' COMMENTS:

Reviewer #1 (Remarks to the Author):

The revised version of the manuscript has certainly improved in several respects, but there are two issues that remain:

1) The nexus fermions in graphite are indeed the joint of four nodal lines. However, as shown in the lower panel of Fig. 2 in <https://arxiv.org/pdf/1505.03277.pdf>, the Berry phases of these nodal lines are such that they add up to 2π . Hence, if by symmetry these lines are required to fall on top of each other, the Berry phase along the resulting one joint line is 2π and hence the dispersion perpendicular to that line is quadratic. This is the case found in the manuscript for pentacarbon. A spinful version of this type of nexus fermion, stabilized by different crystalline symmetries, is discussed in <https://arxiv.org/pdf/1605.06831.pdf>. Thus, while to my knowledge the spinless case of the latter has not been studied by any other paper so far, the degree of novelty of this type of band crossing is limited as the spinful analogue has been studied in the abovementioned publication.

2) Reviewer 2 pointed at possible accuracy problems when calculating the band gap using the PBE functional. In the reply, using a hybrid functional, the authors locate the transition of the material to a state with the exotic band crossing at the much higher strain value of -2% (compared to -0.1%) and similarly high hydrostatic pressures of ~ 5 GPa. Given that the material is metastable and may be hard to synthesize in the first place, the prediction of exotic band crossings under such rather strong deformations may be of limited practical relevance.

The above two points diminish the possible impact of the findings of this manuscript to an extent that I believe they do not deserve publication in Nature Communications, but are better suited for a more specialized journal.

Reviewer #2 (Remarks to the Author):

I have read carefully the revised manuscript together with the reply letter to the previous Referees reports. The questions raised by the Referees are properly answered, and the revised manuscript becomes much clearer. I recommend to publish the manuscript in the present format in Nature Communications.

Reviewer #1 (Remarks to the Author):

1) The nexus fermions in graphite are indeed the joint of four nodal lines. However, as shown in the lower panel of Fig. 2 in <https://arxiv.org/pdf/1505.03277.pdf>, the Berry phases of these nodal lines are such that they add up to 2π . Hence, if by symmetry these lines are required to fall on top of each other, the Berry phase along the resulting one joint line is 2π and hence the dispersion perpendicular to that line is quadratic. This is the case found in the manuscript for pentacarbon. A spinful version of this type of nexus fermion, stabilized by different crystalline symmetries, is discussed in <https://arxiv.org/pdf/1605.06831.pdf>. Thus, while to my knowledge the spinless case of the latter has not been studied by any other paper so far, the degree of novelty of this type of band crossing is limited as the spinful analogue has been studied in the above mentioned publication.

Response: We agree with the reviewer's comment that if the four Dirac nodal lines discussed in <https://arxiv.org/pdf/1505.03277.pdf> (cited as Ref. 27) for Bernal graphite were somehow required to fall on top of each other, the situation would resemble our case. And the reason that this does *not* happen is precisely because the two systems (Bernal graphite and Pentagon Carbon) have distinct symmetry and topology. In Bernal graphite, each Dirac line has a Z_2 protection, and the arrangement of *four* Dirac lines is compatible with the three-fold rotational symmetry with the -1 line at the center and the three +1 lines at periphery. In contrast, such arrangement (four Dirac lines with Z_2 indices (-1,+1,+1,+1)) can *never* occur in Pentagon Carbon, because it is *not* compatible with the four-fold screw rotational symmetry. Moreover, we emphasize that protection mechanisms of the degeneracy in the two systems are distinct: it is from the nontrivial Z_2 (i.e. π Berry phase) in Bernal graphite; whereas it is from four-fold screw rotational symmetry in Pentagon Carbon.

The reviewer mentions another work <https://arxiv.org/pdf/1605.06831.pdf> (cited as Ref. 22 here), which discusses a spin version of the nexus fermion. We do not think that work will limit the novelty of our current discovery, because, as we stressed, spinful and spinless systems are fundamentally distinct in their symmetry and topology classifications. For example, the time reversal operation $\mathcal{T}^2 = +1$ for spinless system, whereas $\mathcal{T}^2 = -1$ for spinful system (which necessitates Kramer's degeneracy). And rotations for spinful systems would also operate on the spin degree of freedom, requiring a double group symmetry representation, which is distinct from spinless systems. Thus, our

current discovery is fundamentally new. Actually, the reviewer also acknowledges that it “has not been studied by any other paper so far”. Regarding novelty, we also emphasize that the triply-degenerate point is only part of our discovery. Besides the intrinsically fascinating novel crystal structure, we also revealed the isospin-1 fermion point and the inter-connected Weyl loops, which have not been discussed in those previous studies.

In the revised manuscript, have revised the discussion on page 13 to clarify this issue:

“... and triple-degenerate points have also been found in a few materials with strong spin-orbit-coupling. The triply-degenerate points here are distinct from those previous examples in the following aspects: First, due to the negligible SOC, Pentagon Carbon is essentially a spinless system, and as we mentioned at the beginning, spinful and spinless systems belong to different symmetry/topological classes (For example, the time reversal operation has distinct properties between the two cases. And rotations for spinful systems will also operate on the spin degree of freedom, requiring a double group representation, distinct from spinless systems.)”

And we have added the following discussion on page 14:

“We note that the different configurations of degeneracy lines in Bernal graphite and Pentagon Carbon precisely reflect their distinct symmetries: The arrangement of four Dirac lines with Z_2 indices $(-1,+1,+1,+1)$ observed in Bernal graphite is allowed by its three-fold rotational symmetry, but it is not compatible with the four-fold screw-rotation in Pentagon Carbon.”

2) Reviewer 2 pointed at possible accuracy problems when calculating the band gap using the PBE functional. In the reply, using a hybrid functional, the authors locate the transition of the material to a state with the exotic band crossing at the much higher strain value of -2 % (compared to -0.1%) and similarly high hydrostatic pressures of ~5 GPa. Given that the material is metastable and may be hard to synthesize in the first place, the prediction of exotic band crossings under such rather strong deformations may be of limited practical relevance.

Response: First, the hybrid functional result is not universally more accurate than PBE result. Given

that hybrid functional approach could overestimate bandgap shown in previous studies, the value of -2 % is better interpreted as an upper bound. Second, the value -2 % is not large for carbon materials due to their strong bonding. We have calculated the strain-stress curve for Pentagon Carbon (see the following figure), which clearly shows that the material has a linear elastic regime up to $\pm 10\%$ strain. Thus, the -2 % strain can be readily achievable.

The figure for strain-stress curve is added to the Supplementary Information. And we have added a sentence in the main text on page 15 to address this point.

“We also mention that like other carbon allotropes, Pentagon Carbon has excellent mechanical property. From first-principles calculation, it shows a linear elastic regime up to $\pm 10\%$ strain (see Supplementary Fig. 7). Hence the strain-induced phase transitions discussed here should be readily achievable.”